# Virus shedding kinetics and unconventional virulence tradeoffs

**Andrew R. Wargo** [1]*, **Gael Kurath** [2], **Robert J. Scott** [3], **Benjamin Kerr** [3]

**1** Virginia Institute of Marine Science, William & Mary, Gloucester Point, Virginia, United States of America, **2** U.S. Geological Survey, Western Fisheries Research Center, Seattle, Washington, United States of America, **3** Department of Biology, University of Washington, Seattle, Washington, United States of America

* arwargo@vims.edu

**Data Availability Statement:** All relevant data are within the manuscript and its Supporting Information files.

**Funding:** This work was supported by the U.S. Geological Survey Western Fisheries Research Center (GK), the University of Washington (ARW, BK, RJS), and National Science Foundation

## Abstract

Tradeoff theory, which postulates that virulence provides both transmission costs and benefits for pathogens, has become widely adopted by the scientific community. Although theoretical literature exploring virulence-tradeoffs is vast, empirical studies validating various assumptions still remain sparse. In particular, truncation of transmission duration as a cost of virulence has been difficult to quantify with robust controlled *in vivo* studies. We sought to fill this knowledge gap by investigating how transmission rate and duration were associated with virulence for infectious hematopoietic necrosis virus (IHNV) in rainbow trout (*Oncorhynchus mykiss*). Using host mortality to quantify virulence and viral shedding to quantify transmission, we found that IHNV did not conform to classical tradeoff theory. More virulent genotypes of the virus were found to have longer transmission durations due to lower recovery rates of infected hosts, but the relationship was not saturating as assumed by tradeoff theory. Furthermore, the impact of host mortality on limiting transmission duration was minimal and greatly outweighed by recovery. Transmission rate differences between high and low virulence genotypes were also small and inconsistent. Ultimately, more virulent genotypes were found to have the overall fitness advantage, and there was no apparent constraint on the evolution of increased virulence for IHNV. However, using a mathematical model parameterized with experimental data, it was found that host culling resurrected the virulence tradeoff and provided low virulence genotypes with the advantage. Human-induced or natural culling, as well as host population fragmentation, may be some of the mechanisms by which virulence diversity is maintained in nature. This work highlights the importance of considering non-classical virulence tradeoffs.

## Author summary

Whether pathogens evolve to cause more or less disease as they adapt to hosts is a long-standing question. Answering this question is a critical step in disease management because it allows for assessment of which emergent pathogens are likely to have devastating health impacts and which are not. Theory ascertains that pathogens should evolve to cause intermediate levels of disease, i.e. virulence, to their hosts. However, this theory is

Ecology of Infectious Diseases grant 0812603 (BK, GK, ARW, RJS). The funders had no role in study design, data collection and analysis, decision to publish, or preparation of the manuscript.

**Competing interests:** The authors have declared that no competing interests exist.

based on limited empirical evidence. We sought to evaluate various assumptions inherent in this theory. To do so we quantified how pathogen transmission (measured as viral shedding) is associated with virulence (measured as host mortality), in a virus that heavily impacts salmon aquaculture and conservation. We found that there were both fitness costs and benefits of virulence for the virus, but the benefits typically outweighed the costs, resulting in selection for increased virus virulence. We did, however, identify scenarios where less virulent virus strains could have the evolutionary advantage. This work provides important insights into virulence evolution and how it might be managed so as to reduce long-term disease impacts.

## Introduction

Virulence tradeoff theory is one of the leading explanations for how pathogen virulence evolves [1]. The theory assumes that virulence comes with both fitness costs and benefits for pathogens, which ultimately results in the evolution of intermediate levels of virulence. In the classical framework, the benefit of virulence is hypothesized to be a positive association with pathogen transmission rate. In contrast, the cost of virulence is hypothesized to be truncation of transmission duration, due to shortening of the host lifespan. This ultimately creates a hump-shaped relationship between virulence and fitness, where pathogens with the highest fitness have intermediate virulence. Here, virulence is defined as host mortality, which is the key driver of the classical tradeoff [2,3]. Pathogen fitness is defined as overall transmission, which is the product of transmission rate and transmission duration.

Empirical evidence of the virulence tradeoff first came from field and laboratory studies of myxoma virus in rabbits [4]. In this system, the tradeoff was originally described as being between virulence and recovery, where decreases in transmission duration due to host death were counter-balanced with increases due to slower recovery. Here, recovery is defined as the rate at which infected hosts, which survive, become no longer infectious. Later, evidence of the transmission rate versus transmission duration tradeoff was presented, which improved fit to the myxoma field data and became the classical framework [5]. The key to the theory is that the relationship with virulence and recovery or transmission rate is a saturating function (typically convex for recovery and concave for transmission rate), allowing for fitness (traditionally $R_0$ –or the pathogen's basic reproduction ratio [6]) to be maximized at intermediate levels of virulence [1]. A plethora of theoretical papers expanded upon virulence tradeoff theory over the subsequent few decades [1,3,7]. However, further empirical evidence of the phenomenon remained sparse, leading many to challenge the framework [8–10].

In recent years, there has been some expansion of empirical studies examining various assumptions inherent to virulence tradeoff theory [2]. There is now a substantial body of literature demonstrating the positive association between virulence and pathogen transmission rate [11–19], some of which demonstrates a concave relationship [20–23]. There is also some evidence that virulence comes with the pathogen benefit of infected hosts having lower recovery rates [24–26]. However, empirical evidence for the cost of virulence is more limited, i.e. saturation between virulence and overall transmission [20,27]. Inherent in tradeoff theory is the assumption that the cost of virulence is caused by the truncation of pathogen transmission duration due to host mortality. Outside the myxoma system [5,28], few studies have empirically measured such a relationship and there are a variety of scenarios where it may not hold; for example, if transmission largely occurs before host mortality begins [29]. A major hurdle to validating this assumption has been quantification of pathogen transmission [30]. Where

transmission has been quantified, it is typically only done so at a few discrete time points [31], which may not provide a complete representation of overall transmission, or other epidemiologically important parameters such as $R_0$. Other metrics, such as pathogen loads within the host, have thus been used to estimate transmission [2,30]; however, in many cases, their association with overall pathogen fitness has not been well-established. Virulence can also be a challenging metric to quantify, particularly in systems where host mortality is very slow or uncommon [32], or unethical to measure [33]. Again, proxies for virulence have been developed, but their associations not fully validated [2,34]. Many of these associations are likely to be complex, and not necessarily linear as assumed, which could have confounding effects on virulence tradeoff associations [35]. Despite these limitations, there is increasing evidence that pathogen strains of intermediate virulence tend to be those that are dominant and have highest fitness at a population level [27,36,37]. However, there are notable exceptions [38], and the relationship between fitness and virulence is likely to be system dependent [8,39,40]. Furthermore, selection for intermediate virulence types could occur through other mechanisms besides a classical virulence tradeoff [9,41–44]. Ultimately, understanding the trajectory and drivers of virulence evolution is a critical component to disease management, particularly for emerging pathogens [45–47].

Here, we sought to quantify the relationship between fitness and virulence for infectious hematopoietic necrosis virus (IHNV) in rainbow trout (*Oncorhynchus mykiss*). In particular, we aimed to test the cost-benefit assumption of tradeoff theory by examining how virulence was associated with the fitness traits of transmission rate and transmission duration. To quantify transmission, we measured viral shedding over the entire infectious period. This allowed for a continuous parameterization of transmission, rather than being constrained to incomplete estimation of host-to-host transmission at discrete time points.

Infectious hematopoietic necrosis virus is a negative sense single stranded RNA virus in the Rhabdoviridae family [48]. The virus is endemic in salmonid fishes of the Pacific Northwest of North America and has spread worldwide through aquaculture [49–51]. It is particularly problematic in rainbow trout, in which it causes an acute disease leading to necrosis of the kidney and spleen and ultimately death [52]. Epidemics reaching over 50% mortality are not uncommon, and losses to salmonid aquaculture, conservation, and fisheries are significant [53–57]. Disease severity is shaped by a variety of host, virus, and environmental factors [58–60]. The virus demonstrates a substantial degree of genetic diversity, particularly in rainbow trout aquaculture, with viral genotypes also displaying a wide range of virulence levels [56,58,59,61–63]. Previous studies have indicated that virulence is positively associated with infectivity, peak-viral load, in-host competition, and early shedding in rainbow trout [19,64–66]. There is also evidence that virulence was correlated with field displacement events in steelhead trout (anadromous *Oncorhynchus mykiss*), but not with other viral fitness indicators measured in laboratory studies [67,68]. The virus is believed to be primarily transmitted horizontally through water with viral exit via body fluids (urine, feces, mucus, milt, ovarian fluid) and viral entry through gills and fin bases [69–75]. Vertical transmission has been eliminated in aquaculture with egg disinfection, but likely occurs in the wild [76–79]. Dose response studies have characterized the relationship between viral concentration in the water and infectivity [68,80–82]. As such, viral shedding quantification has been established as a proxy for transmission [82–86].

Various approaches have been used to manage the virus [51,56,87]. Increased biosecurity (sanitation, disinfection, quarantine) is the mostly widely implemented [49,88]. Vaccines are also available, but are used on a limited basis due to their high production and delivery costs [89–91]. Another widely practiced strategy is fish culling, where entire raceways or ponds holding fish at an aquaculture or hatchery facility are euthanized and tanks disinfected when a disease outbreak is suspected and mortality reaches a prescribed threshold [49,55,57]. The goal

of culling is to remove infected hosts and prevent spread of the virus to other fish cohorts at the facility, and has proven effective in some cases [92].

We conducted three *in vivo* experimental studies of IHNV in rainbow trout comparing the virulence and fitness of three pairs of viral genotypes. Previous studies have indicated that viral fitness differences are most visible when genotypes are compared side-by-side in a coinfection design [93], and coinfection could have important consequences for the evolution of virulence [11]. As such we examined IHNV fitness and virulence in single and mixed infections (one genotype pair per experiment), using the same dosage of each virus genotype for both infection types. Virulence was quantified using daily fish mortality and fitness quantified via viral shedding from individual fish over the 30-day course of infection. The associations between virulence and fitness were elucidated to determine how they conformed to classical virulence tradeoff theory. A population level transmission model was constructed and parameterized with experimental data, to infer the trajectory of virulence evolution in a trout farm environment. The impacts of host culling on virulence evolution were also investigated.

## Results

### Survival

In general, fish mortality began around day 5 in all experiments, and remained elevated until day 15, after which it tapered (Fig 1). In experiments 2 and 3, there was a second wave of mortality from day 25–30, primarily in fish exposed to IHNV genotype LR80. A cox proportional hazard analysis revealed that the hazard of death was significantly lower for fish infected with genotypes LV and MER95 compared to HV and LR80, in experiments 1 (Fig 1A, hazard ratio (HR) LV/HV = 0.21 +/- 0.48 (1 standard error: SE), log rank (LR) = 18.7 on 2 degrees of freedom (df), P<0.001) and 2 (Fig 1B, HR MER95/LR80 = 0.27 +/- 0.53, LR = 15.7 on 2 df, P<0.001) respectively. As such, HV and LR80 were the more virulent IHNV genotypes in these two experiments. There was no significant difference in the hazard of death with fish coinfected with both genotypes, compared to infected singly with the more virulent genotype, for either experiment (p>0.1). In experiment 3, there was no significant difference in the hazard of death of fish infected with genotype LR80 compared to HV (Fig 1C, HR LR80/HV = 0.89 +/- 0.36, LR = 0.11 on 1 df, p = 0.70), despite a suggestive trend of more rapid mortality for genotype HV. Mixed infections were not assessed in experiment 3. No fish died in any of the mock exposed control groups. Our results thus indicated that virulence differed by the most between genotypes in pair 1 (cumulative mortality 30% for LV and 85% for HV), more

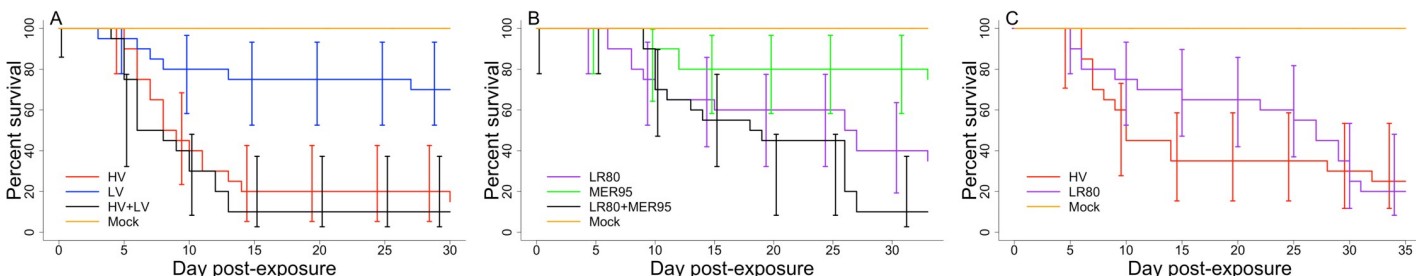

**Fig 1. Cumulative survival of fish exposed to IHNV.** Panels show Kaplan-Meier survival curves for fish exposed to IHNV genotypes HV (red), LV (blue), MER95 (green), LR80 (purple), mixed infections (black, 1:1 ratio of two genotypes in experiment), or a mock negative control (orange) in experiments 1–3 (A-C, respectively). Mixed infections were not assessed in experiment 3. Error bars show 95% confidence interval based on cumulative hazard, derived from Kaplan-Meier analyses. All treatments began with 20 fish, with the exception of mock control which had 8 fish. Mortalities were recorded daily for the entire course of the experiment (30–35 days).

moderately for pair 2 (cumulative mortality MER95: 25%, LR80: 65%), and by the least for pair 3 (cumulative mortality LR80: 80%, HV: 75%).

## Number of fish shedding

In all experiments the number of fish shedding rapidly increased such that all fish began shedding by days 1 or 2 post exposure to virus (Fig 2 and Fig A in S1 Text). Shedding then gradually decreased and by day 15 the majority of remaining live fish had stopped shedding. However, a small number of live fish (1–5), were still shedding detectable virus at the end of each experiment (day 30–35). It is worth noting that some fish continued to shed detectable viral RNA for several days after fish death (Fig A in S1 Text). In general, this was more common for the higher virulence genotypes (HV and LR80) compared to the lower virulence ones (LV and MER95). Previous studies indicate that virus shed from dead fish loses infectivity rapidly after the time of host death [83,94]. As such, virus shed after 24 hours post-mortem was excluded from the analyses. The data analyses indicated that the probability of fish shedding virus decreased through time for both experiment 1 and 2 (Tables A and B in S1 Text; day main effect; Exp. 1: Coef. -0.27 +/- 0.07, $Z_{1,927}$ = -3.9, P<0.001; Exp. 2: Coef. -0.14 +/- 0.03, $Z_{1,1214}$ = -4.9, P<0.001), with a more rapid decline in the probability of shedding for the less virulent genotype (LV Exp. 1, MER95 Exp. 2) compared to the higher virulent genotype (day * genotype interaction; Exp. 1: Coef. -0.31 +/- 0.08, $Z_{1,927}$ = -3.9, P<0.001; Exp. 2: Coef. -0.24 +/- 0.05, $Z_{1,1214}$ = -5.4, P<0.001). However, the initial probability of shedding on day 1 did not significantly differ between the two genotypes in either experiment (genotype main effect; Exp. 1: $Z_{1,927}$ = -0.36, P = 0.7; Exp. 2: $Z_{1,1214}$ = -1.8, P = 0.08). No competition effect was found, indicating that the probability of shedding did not differ between single and mixed infections for any virus genotypes within experiment 1 or 2. This was despite a suggestive trend of a reduction in the number of fish shedding for mixed infections compared to single infections in experiment 2 (Fig 2B). For experiment 3 the data analysis indicated that there was a significant reduction in the probability of viral shedding through time (Table C in S1 Text; day main effect; Coef. -0.76 +/- 0.15, $Z_{1,428}$ = -5.2, P<0.001). However, the initial and rate of change in the probability of shedding through time, did not differ between genotypes HV and LR80 (P>0.05).

## Shedding intensity

In general, the mean amount of virus shed per day from those fish actively shedding (i.e. shedding intensity) peaked by day 2–3, then rapidly declined 2–3 orders of magnitude until day 3–5 (Fig 3). Afterwards shedding intensity remained stable, although the number of live fish

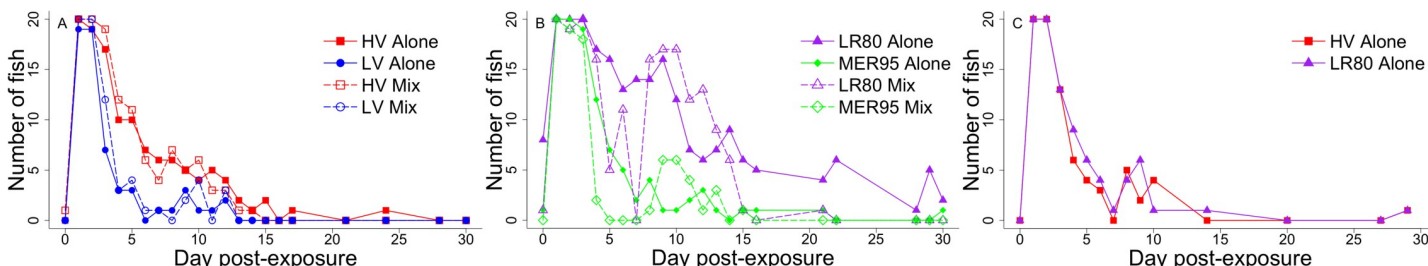

**Fig 2. Number of fish shedding virus.** Panels show number of remaining live fish shedding detectable quantities of IHNV RNA for genotypes HV (red), LV (blue), MER95 (green), LR80 (purple), in single (solid lines) or mixed infections (dashed lines), in experiments 1–3 (A-C, respectively). Mixed infections were not assessed in experiment 3. There were 20 total fish per treatment group. Symbols indicate time points where water samples were taken for shedding detection, measured by qPCR. Fish were excluded from the day after death onwards.

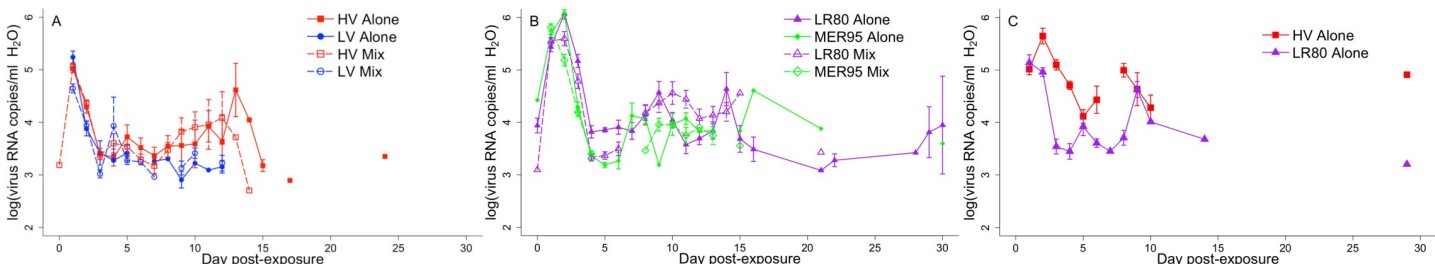

**Fig 3. Virus shedding intensity.** Panels show the mean amount of IHNV [log10(virus RNA copies/ml $H_2O$)] shed per day for genotypes HV (red squares), LV (blue circles), LR80 (purple triangles), and MER95 (green diamonds), in experiments 1 (A), 2 (B), and 3 (C). Points represent days samples were taken to quantify viral shedding in single (closed symbols and solid lines) and mixed infections (open symbols and dotted lines). Error bars show +/- 1 standard error of the mean. Only live fish shedding detectable virus into the water were included in the figure. A missing point indicates no virus shedding was detected by qPCR for all fish in a given treatment. Points not connected by a line indicate virus dropped below detection at sampling time points in-between. Points without standard error bars indicated fewer than 2 fish were positive for virus. Number of fish included in the mean values for each data point are the number of live fish shedding at the same time point in Fig 2.

shedding continued to decrease. Analysis of experiment 1 data indicated that there was a significant decrease in viral shedding intensity through time (Fig 3 and Table D in S1 Text; day main effect; Coef. -0.09 +/- 0.02, $T_{1, 304}$ = -5.53, P< 0.001), but the rate at which shedding intensity decreased did not differ between the genotypes or in single versus mixed infections (day*genotype and day*competition interactions; P>0.1). On average, genotype LV in competition was found to have a lower shedding intensity than genotype HV alone and in competition (genotype*competition interaction; Coef. -0.34 +/- 0.17, $T_{1, 304}$ = -2.0, P = 0.04). However, there was no significant difference in shedding intensity between LV alone versus in competition or HV alone (lsmean, P>0.05). Thus, this suggests some level of competition effect, where the shedding of the less virulent genotype (LV) was suppressed to a slightly larger degree than the high virulent genotype (HV), such that the more virulent genotype had the shedding advantage when in direct competition.

For experiment 2 the data analysis indicated that viral shedding intensity decreased through time (Table E in S1 Text; day main effect; Coef. -0.05 +/- 0.01, $T_{1,520}$ = -5.40, P< 0.001), with the rate of decrease being faster for mixed infections compared to single infections (day*competition interaction; Coef. -0.05 +/- 0.02, $T_{1,520}$ = -2.58, P = 0.01), as well as faster for genotype MER95 compared to genotype LR80 (day*genotype interaction; Coef. -0.03 +/- 0.01, $T_{1,520}$ = -2.04, P = 0.04). The model also indicated, that on average, the shedding intensity for MER95 was lower than LR80 in mixed infections, but not single infections (genotype*competition interaction; Coef. -0.47 +/- 0.18, $T_{1,520}$ = -2.62, P = 0.01). As for experiment 1, this indicated a competition effect wherein the more virulent genotype maintained a shedding advantage in mixed infections.

For experiment 3 the data analysis indicated that there was a significant decrease in shedding intensity through time (Table F in S1 Text; Coef. -0.07 +/- 0.01, $T_{1,280}$ = -7.40, P< 0.001). However, there was no difference in the rate of this decrease or overall shedding intensity between the two viral genotypes (P>0.1).

## Total virus shed

In an effort to estimate overall viral fitness, the total amount of virus shed by each fish over the course of infection was compared between genotypes in single and mixed infections, in each experiment. Only the quantity of virus shed by a fish before 24 hours after the time of death was included in the total. In general, 1–3 orders of magnitude more virus was shed during the initial peak period (defined here as days 0–4) than during the post peak period (days 5 onward) (Fig 4). Thus, these shedding periods were analyzed separately, so as to more fully capture the

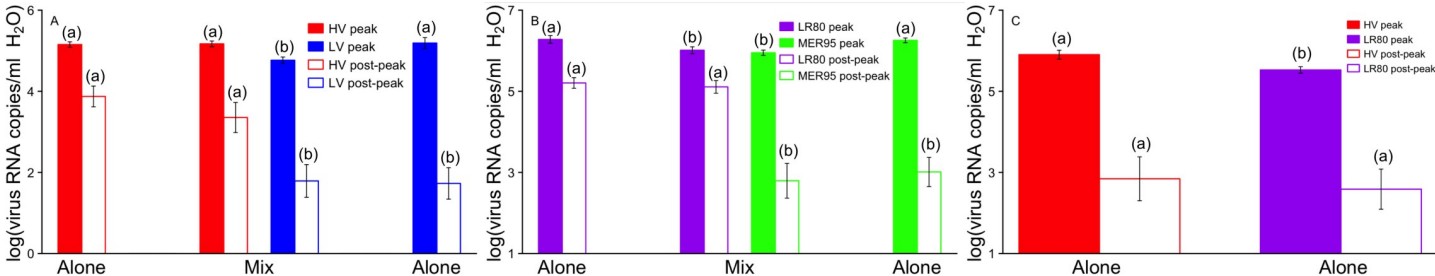

**Fig 4. Total virus shed.** Panels show mean total virus shed [log10(total virus RNA copies/ml $H_2O$+1)] for the peak (days 0–4, filed bars) and post peak periods (days 5 onwards, open bars) for genotypes HV (red), LV (blue), LR80 (purple), and MER95 (green), in experiments 1 (A), 2 (B), and 3 (C). Single infections are shown on the panel edges and mixed infections together in the panel center, to denote that viral quantities for the two genotypes within an experiment in mixed infections came from the same fish. Error bars show +/- 1 standard error of mean. Shedding from all fish is included in the mean up to one day after fish death. For experiment 2, days 7 and 16 were excluded from total because only single infection samples were taken. Letters above bars (a and b) denote significant differences (P<0.5, see text) between bars within panels and infection periods. Statistical comparisons were not made across panels (experiments) or infection periods (peak vs. post-peak).

kinetics of viral shedding. During the peak period, for experiment 1 (Fig 4A) the analysis revealed that total shedding of the less virulent genotype (LV) was lower in mixed compared to single infections (Table G in S1 Text; genotype* competition interaction; Coef. 0.44 +/- 0.15, $T_{1,18}$ = 3.05, P = 0.01). Genotype LV also shed less than the high virulent genotype (HV) in mixed infections (lsmeans, Coef. 0.40 +/- 0.05, $T_{1,18}$ = 8.15, P < 0.001). There was no difference in total shedding between the two genotypes in single infections (P>0.05). There was also no effect of competition on the total amount of virus shed for genotype HV (P>0.05). In other words, genotype LV experienced competitive suppression but genotype HV did not. This resulted in genotype LV shedding less total virus in mixed infection than HV did alone (lsmeans, Coef. 0.39 +/- 0.14, $T_{1,18}$ = 2.8, P = 0.0496). In experiment 2 (Fig 4B), the analysis revealed that both genotypes shed less total virus in mixed compared to single infections (Table H in S1 Text, Coef. 0.29 +/- 0.09, $T_{1,58}$ = -3.28, P = 0.002), indicating that both genotypes experienced competitive suppression. However, there was no significant difference in total shedding between the genotypes (P>0.05). For experiment 3 (Fig 4C) the analysis revealed that genotype LR80 shed significantly less virus than genotype HV (Table I in S1 Text, Coef. -0.38 +/- 0.14, $T_{1,38}$ = -2.78, P = 0.008).

During the post peak period, the analyses revealed that the less virulent genotypes (LV and MER95) shed less total virus than the more virulent genotypes (HV and LR80), in experiment 1 (Fig 4A and Table J in S1 Text, Coef. -1.95 +/- 0.33, $T_{1,71}$ = -6, P < 0.001) and 2 (Fig 4B and Table K in S1 Text, Coef. -3.07 +/- 0.26, $T_{1,74}$ = -11.8, P < 0.001) respectively. There was no effect of competition on the total shedding of either genotype during the post peak period, for either experiment 1 or 2 (P>0.05). For experiment 3 the analysis indicated that there was no significant difference in the total amount of virus shed between the two genotypes, during the post peak period (Fig 4C and Table L in S1 Text, Coef. -0.60 +/- 0.33, $T_{1,35}$ = -1.82, P = 0.07).

## Association viral fitness and virulence for individual fish

The relationship between IHNV fitness parameters (transmission rate, recovery rate, total transmission potential) and virulence (1/day of death) was investigated on an individual fish basis (Fig 5). Transmission rate was defined as peak viral load shed, recovery rate as 1/duration of shedding, and transmission potential as total amount of virus shed. Various functional forms (linear, quadratic, exponential) were investigated to determine the associations between the parameters. None of these functions provided a better fit than a standard uniform function for any of the fitness parameters (transmission rate: $T_{1,119}$ = 99.83, P<0.001; recovery rate:

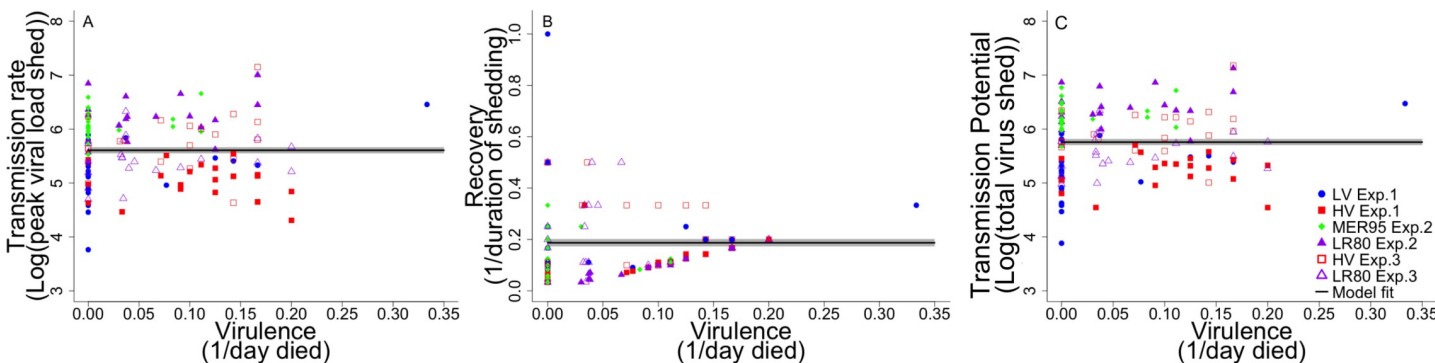

**Fig 5. Association between virulence and fitness.** Panels show association between (A) transmission rate (peak viral load shed [log10(virus RNA copies/ml $H_2O$)]), (B) recovery rate (1/duration of shedding (days)) and (C) total transmission potential (total virus shed [log10(virus RNA copies/ml $H_2O$)]), with virulence (1/day fish died). Each data point represents an individual fish infected with a given IHNV genotype in a given experiment (see in figure legend). Only fish exposed to a single virus genotype, which shed detectable virus, are shown. Mixed infections are excluded due to potential confounding effects of genotype interactions on virulence. The solid line represents the best fit ((A) Y = 5.607+/-0.057, (B) Y = 0.187 +/- 0.014, (C)Y = 5. 758 +/-0.056) to the data +/- 1 standard error (SE) (grey area). The line demonstrates a uniform, non-significant association in all cases. As such, virulence explained none of the variance in the viral fitness parameters on an individual fish basis.

$T_{1,119}$ = 13.34, P<0.001; total shedding: $T_{1,119}$ = 99.83, P<0.001). As such, virulence measured as day of fish death, did not explain any of the variance in the viral fitness parameters on an individual fish basis. Ultimately, fish-to-fish variation in fitness was high for each level of virulence, particularly for fish that did not die (i.e. virulence equal zero).

## Transmission modelling

A transmission model, tracking the number of susceptible, infected, recovered, and dead fish (SIRD) was constructed (Eqs 1–4). The model was parameterized from experiments conducted here and a typical aquaculture setting to elucidate under what conditions high or low virulence genotypes might become dominant in the field. The model indicated that when the high (HV) and low (LV) virulence IHNV genotypes simultaneously infected a fish population, the high virulence genotype became dominant and the low virulence genotype was driven to extinction (Fig 6A). However, the low virulence genotype had a higher peak incidence, allowing it to infect a greater number of cumulative fish until day 200. This appeared to be driven by a slightly higher mean transmission rate for the low virulence genotype (Table 1). When fish populations were only infected with a single IHNV genotype, the high virulence genotype again became dominant. However, both genotypes persisted in their respective populations and the cumulative number of fish infected was only slightly higher for the more virulent genotype (Fig 6C). When culling was implemented, if both genotypes simultaneously infected the fish population, all fish were culled and both genotypes were driven to extinction at the same time (Fig 6B). It is interesting to note that, in this scenario, the low virulence genotype infected a larger cumulative number of fish, again likely driven by a slightly higher transmission rate. When fish populations were only infected with a single IHNV genotype, culling the high virulence genotype occurred very rapidly (day 10), whereas the low virulence genotype persisted for two thirds of the fish production cycle (250 out of 365 days) (Fig 6D). This allowed the lower virulence genotype to infect a larger cumulative number of fish very early during the epidemic, as well as maintain a much longer epidemic. Overall, in the absence of culling high virulence was selected, largely due to competition between genotypes in the same sub-population. In the presence of culling, low virulence was selected, primarily due to effects of genotypes on culling in different subpopulations. In all scenarios the cumulative number of fish infected by the two genotypes was within an order of magnitude.

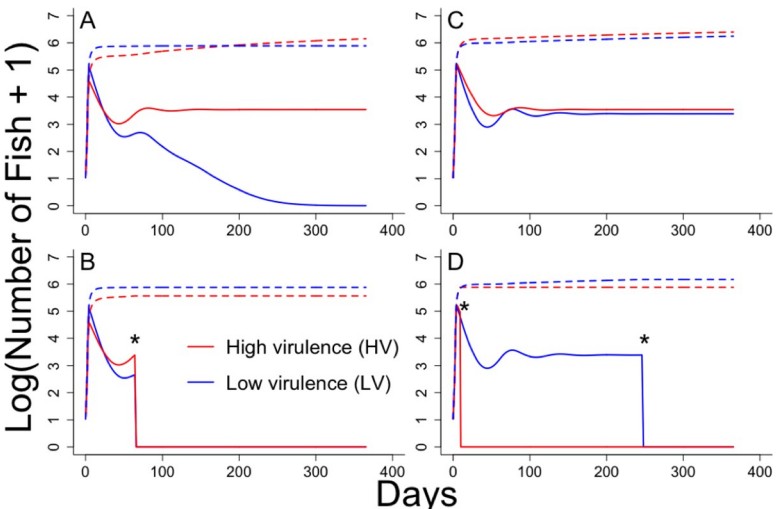

**Fig 6. Model output of IHNV transmission at a population level.** Plots show incidence (solid line) and cumulative total (dashed line) number of fish (Log + 1) infected with either the high virulence (HV–red) or low virulence (LV–blue) IHNV genotype, through time (days). Data are derived from an SIRD (susceptible, infectious, recovered, dead) model (see methods Eqs 1–4) parameterized from experimental data (see Table 1). For panels A and B, populations of fish are simultaneously infected with both genotypes HV and LV, and thus both lines represent the same fish population. For panels C and D, populations of fish are singly infected with each genotype separately, and thus each line represents a different fish population. For panels B and D, culling is implemented when mortality reaches 30% of the initial fish population size. Culling involves removal of all fish from the population, regardless of infection status (i.e. set S, I, R, and D to zero). Culling events are denoted by * on the figure. State variables are initially set to S = 200,000, infected HV = 10, infected LV = 10, R = 0, D = 0; to reflect conditions in a raceway on a standard trout farm (Table 1).

## Discussion

We investigated the association between virulence and fitness for IHNV in rainbow trout. We then assessed whether these associations conformed to classical virulence tradeoff theory [1]. Although virulence covaried with other traits, our observations did not conform to conventional theory. Firstly, we observed that, on average, fish infected with higher virulence IHNV genotypes shed for longer than those infected with lower virulence genotypes. In other words, the recovery rate (1/duration of shedding) was higher for low virulence genotypes. This was true for two out of three of the genotype pairs investigated, where the difference in virulence between genotypes was large (HV vs. LV) or moderate (LR80 vs. MER95). For the third pair, there was no significant difference in virulence or shedding duration between the genotypes.

Surprisingly, the more virulent genotypes shed for longer, despite killing fish more quickly. For example, in experiment 2 by day 10 approximately 90% of fish infected with the low

**Table 1. Parameter values SIRD transmission model.**

| Symbol (HV, LV)* | Definition (units) | Value (HV, LV)* | Source |
|---|---|---|---|
| $\varphi$ | Fish stocking rate (fish/day) | 548 (entire population/year) | [134,135] |
| $\beta_{HV}, \beta_{LV}$ | Transmission rate (fish infected/day) | $1.802 \times 10^{-5}$, $2.085 \times 10^{-5}$ | Experiment 1 data [80], field parameters [134–137] |
| $\rho_{HV}, \rho_{LV}$ | Recovery rate (fish recovered/day) | 0.069, 0.171 | Experiment 1 data |
| $\alpha_{HV}, \alpha_{LV}$ | Death rate due to infection (fish dead/day) | 0.079, 0.041 | Experiment 1 data |
| $\gamma$ | Fish harvest rate (fish/day) | $2.7 \times 10^{-3}$ (entire population/year) | [135] |

* Symbols and values are given for IHNV genotypes (*g*) HV and LV respectively. If single value given, same for both genotypes.

virulence genotype were alive and only 5% were shedding, whereas only 70% infected with the high virulence genotype were alive, all of which were shedding. Overall, the association between shedding duration and host lifespan was weak, contradictory to virulence tradeoff theory that infected hosts which live for longer will also transmit for longer. Instead, shedding duration was a function of host clearance, with the low virulence genotypes being more rapidly cleared than the high virulence genotypes. These results indicate that the immune system more effectively controls infection with low virulence IHNV genotypes compared to high virulence ones, and is consistent with previous findings in this system [95–97]. Such a phenomenon was postulated for the myxoma virus system, which serves as the foundation for classical virulence tradeoff theory [27,28,98]. However, our data differed in that the relationship between virulence and recovery was curvilinear for the myxoma system and not here for IHNV [4]. As such, our results indicate that transmission duration of IHNV is likely to continue to increase with virulence. There was no evidence that reductions in recovery rate are outweighed by truncation of host lifespan at the virulence levels studied here. This is likely because shedding of the virus is very acute and mostly occurs before mortality begins [82–84]. It is important to note that we excluded viral shedding after fish death, despite some fish continuing to shed viral RNA post-mortem. However, this phenomenon was observed most frequently for the higher virulence genotypes, and as such, if included, shedding from dead fish would only have further increased the transmission duration advantage of virulence. There are limited studies on the transmission of IHNV from dead fish, which indicate infectivity declines within 1–2 days [83,94], but this warrants further exploration.

Secondly, we did not observe a consistent relationship between transmission rate and virulence. There was no significant difference in the peak probability or intensity of shedding between high and low virulence genotypes. The exception to this was in mixed infections, where there was an indication that the high virulence genotype had a shedding intensity advantage, but this effect was small and inconsistent. Again, these patterns were observed for the two genotype pairs where virulence was most disparate, but not the third where virulence and shedding intensity were similar. There was some evidence that shedding intensity decreased more rapidly through time for the less virulent genotype in one of the genotype pairs. As such, shedding differences between genotypes were more pronounced in the later part of infection. This finding further supported the hypothesis that host clearance is more effective for low virulence genotypes. Overall, these results did not align with tradeoff theory which predicts a curvilinear relationship between transmission rate (here estimated as shedding rate) and virulence, with more virulent genotypes having the advantage [5]. In contrast, we estimated that the less virulent genotype in general had equivalent, and in one case (LV vs. HV) slightly higher peak transmission rates, although the difference was not statistically significant.

Collectively, the associations between virulence, shedding duration, and shedding rate resulted in the high virulence genotype generally having the overall fitness advantage. This was observed when quantifying total shedding and was most apparent during the post peak period of infection (Fig 4). Furthermore, parameterization of a transmission model with the experimental data indicated that high virulence would be selected within a population of hosts in the long-term. Interestingly, this was only apparent late into the epidemic and surprisingly the low virulence genotype reached a higher peak incidence when both genotypes infected the population simultaneously. Ultimately, the transmission model indicated that the benefit of the high virulence genotype was maintained but less pronounced when comparing cumulative infection levels rather than incidence, and largely driven by the lower recovery rate rather than transmission rates. It is important to note that the transmission model was parameterized with mean values for each genotype, regardless of whether or not they were significantly different.

How the variance around this mean impacts the evolutionary outcome is worthy of exploration, but beyond the scope of the present work, which intended for the models to be used for inference rather than deterministic prediction.

We extended the transmission model to investigate how culling might impact virulence evolution. In this case, culling resurrected a virulence tradeoff for IHNV. Because culling was triggered by the specified level of host mortality, fish in a population infected with the high virulence genotype were more likely to be culled than those infected with the low virulence genotype. This disproportionally truncated the transmission duration of the high virulence genotype, creating a mortality cost to high virulence. This resulted in a scenario where the low virulence genotype had the overall advantage, despite higher recovery rates. Clearly, the level of mortality at which culling is triggered would have an impact on the fitness of genotypes. But in theory, genotypes should evolve to reach mortality rates that fall just under the culling threshold to maximize fitness. This implies that culling might be a tool to curtail the evolution of increased virulence and warrants further investigation. It would also be interesting to explore how natural culling, for example through diseased animals being more susceptible to predation, might impact virulence evolution. For influenza virus it was theorized that culling would result in the evolution of increased virulence [99]. However, in that system transmission rate was found to have a larger impact on $R_0$ than transmission duration, whereas here we found the opposite. A more generalized framework postulated that culling would select for decreased virulence due to reductions in host density [100]. These disparate predictions further highlight that virulence evolution is likely to be driven by a variety of system-specific host and pathogen factors.

The transmission model also indicated that whether or not fish populations (here modelled as raceways) on a farm are infected with a single or multiple IHNV genotypes, may impact the evolutionary outcome of virulence. This was most pronounced in the presence of culling, wherein multiple genotype infections in the population resulted in complete eradication of both genotypes at the same time point. This was because culling of the low virulence genotype was triggered by mortality caused by the high virulence genotype. Interestingly, even in this case the low virulence genotype infected a larger total number of fish due to its slightly higher transmission rates and peak incidence. Under this coinfection scenario the virus might evolve to increase initial shedding rates in an effort to maximize transmission before culling occurs, as postulated by tragedy of the commons theory [101,102]. How this might influence the evolution of virulence of IHNV is unclear, given that peak shedding does not appear to be correlated with virulence for IHNV. In the absence of culling, lack of multiple infections allowed for the persistence of both the high and low virulence genotypes, despite a small advantage for the high virulence genotype. If virus migration between fish raceways and populations is high the ability of multiple genotypes to co-exist might become diminished because of an increase in concurrent infections. As such, our results indicate that population structure or fragmentation could be a mechanism by which virulence diversity is maintained [103–105].

Information on the prevalence of mixed IHNV genotype infections in field populations is limited. The data available indicate multiple genotypes can co-circulate in populations, but typically epidemics involve a single dominant genotype [106,107]. This may be because prior infection with a single IHNV genotype quickly reduces the susceptibility of hosts to a second infection [66,67]. As such, mixed genotype infections in a single fish are believed to be uncommon in the field, although they are easily established in laboratory infections [64,108]. In general, we found that mixed infections largely mirrored the dynamics of single infections in the present study. However, there was some evidence of competitive suppression of the less virulent genotype, which would provide high virulence types with an even greater advantage if mixed infections do occur in nature. It is unclear what mechanisms might be driving

competition between IHNV genotypes. It was observed that the level of mortality in mixed infections was not significantly greater than that observed for single infections of the high virulence genotype, suggesting there may not be resource limitation [109]. Competitive suppression was most pronounced during the peak infection period, so it is possible that it may be innate immune mediated [95,96]. This is in-line with the observation that less virulent genotypes were cleared more quickly given that they also suffered more competitive suppression.

An interesting finding of our study was that virulence-fitness associations were often weak at an individual fish level. Tradeoff theory hypothesizes that curvilinear relationships exist between transmission rate and virulence, as well as recovery rate and virulence. This is believed to result in overall pathogen transmission (here estimated as total shedding) being maximized at intermediate levels of virulence. We did not observe any such associations between the three viral fitness traits (transmission rate, recovery rate, and total transmission potential) and virulence, on an individual fish basis (Fig 5). As such, the individual fish data did not conform to classical tradeoff theory. There were suggestive trends that the fish with the least viral shedding had the lowest virulence and the fish with the highest shedding had intermediate virulence. However, this relationship was overwhelmed by fish-to-fish variation. The association between fitness and virulence is not typically examined on an individual host basis, but this comparison revealed that variation in host susceptibility may be an equally important driver of virulence evolution as viral genotype. This could provide another mechanism by which variation in virulence can be maintained in the field. At a population level, mean differences in fitness between genotypes should direct their evolution, however, high levels of between host variation could slow this process, or even disentangle it [110]. There is increasing evidence that host heterogeneity in susceptibility could be a major and often overlooked driving force in virulence evolution [111]. This can have particularly pertinent implications for disease control, where heterogeneity in susceptibility could lead to inaccurate estimates of efficacy [81,112].

Ultimately, our results did not conform with classical virulence tradeoff theory, in that the traditional transmission duration truncation cost and greater transmission rate benefit to high virulence, were not observed. We observed instead that high virulence genotypes had longer transmission durations and low virulence genotypes did not have consistently lower transmission rates. This did provide some evidence of a tradeoff, and may be a mechanism by which IHNV diversity is maintained in the field. However, contrary to classical theory, infected host lifespan had very little effect on transmission and did not drive the virulence tradeoff. We certainly cannot rule out the possibility that other traits not measured here, such as persistence or environmental stability [113–115], may demonstrate other costs and benefits for high and low virulence IHNV genotypes, or that sub-lethal effects of virulence might impact fitness. However, these would fall outside the classical tradeoff of reduced transmission due to host mortality. As with any such study, different outcomes might also be observed under different environmental conditions. For example, mortality levels might be higher or lower in the field compared than those we observed in the laboratory. Previous studies indicate this is unlikely to switch the ranking of virulence between the IHNV genotypes studied here [58,67,68], but cannot be ruled out completely. If this were to occur it would indicate that the virulence tradeoff is environmentally dependent and might not be the only driving force for virulence evolution in the field, where environmental variation can be high. Likewise, it is plausible that different patterns would be observed with other IHNV genotypes. Here we studied three IHNV genotype pairs, which exhibited a range of virulence levels. It may be that the high virulence genotype selected did not have a virulence level high enough to show classical costs due to host mortality. It should be noted that at the time these studies were conducted, the virulence of the high virulence HV genotype used here was considered the highest of any genotype observed in the field [116]. This might indicate that the cost of higher virulence is so great,

genotypes exhibiting this phenotype have very low, or even zero, fitness. As far as we are aware, such a rapid, asymmetrical drop-off of fitness with virulence has not been observed in any system, but certainly warrants further investigation. Virulence diversity of IHNV in the field is known to be relatively high [56,58,59,61–63], which again does not exclude the possibility of a classical virulence trade-off, but does suggest there may be other equally important mechanisms by which virulence diversity is maintained.

Collectively, our results indicate that the classical virulence tradeoff may not be the only mechanism driving virulence evolution in all systems, although it is likely important in some [20,37,117,118]. Others have also found as we did that on average, more virulent strains had the fitness advantage [24–26]. This further supports continued studies into the generalizability of this phenomenon in other systems. Why some systems conform to classical virulence-trade-off theory and other do not remains an important research question. In the case of IHNV, the very acute nature of viral shedding may be an important factor. This study also provides empirical evidence that it may be necessary to expand virulence trade-off theory to include non-classical fitness traits [19] as well as consideration of host heterogeneity and population fragmentation, as others have argued [2,119]. Inclusion of anthropogenically induced tradeoffs may also be warranted, particularly where disease management occurs [46,120].

## Methods

### Ethics statement

Fish rearing and experiments were conducted at the U.S. Geological Survey Western Fisheries Research Center in Seattle, Washington and approved by the University of Washington IACUC.

### Virus and host

The viruses used in these studies were infectious hematopoietic necrosis virus (species: *Salmonid novirhabdovirus*) isolates HV (previously 220:90; [64,116]), LV (previously WRAC; [64,121,122]), MER95 [67,107], and LR80 [123]. These isolates were previously typed by the sequence of the middle portion of their glycoprotein gene (mid-G) as mG010M, mG009M, mG111M, and mG007M, respectively [107,124]. Given that each viral isolate had a unique mid-G sequence, herein, they are referred to as genotypes. These genotypes were chosen because previous data indicated they span the range of IHNV virulence in *O. mykiss* [58,68]. Viruses were propagated at a multiplicity of infection of 0.001–0.005 on Epithelioma Papulosum Cyprini (EPC) cells [125] in Eagle's minimum essential media (Gibco) supplemented with 10% fetal bovine serum (MEM-10) to obtain viral stocks stored at -80˚C until use in experiments, as previously described [126]. Titers of virus stocks were determined using highly replicated plaque assays [108,127] as HV: $2.13 \times 10^8$; LV: $3.76 \times 10^7$; MER95: $6.00 \times 10^7$; and LR80: $1.24 \times 10^9$ pfu/ml. Hosts were research grade rainbow trout (*Oncorhynchus mykiss*), averaging 1-3g in size, obtained from Clear Springs Foods, Inc. Fish were reared at 15˚C on sand-filtered and UV-irradiated freshwater, and fed a commercial trout diet (Skretting) at 1–3% body weight per day.

### Experimental design

For each experiment, fish were exposed to $2 \times 10^5$ pfu/ml of one of two virus genotypes (exp. 1 = HV and LV, exp. 2 = LR80 and MER95, exp. 3 = HV and LR80) in 1 L of water using a static immersion challenge batched by treatment, as previously described [83]. As such, each experiment contained single genotype infections of different IHNV genotype pairs. These

pairs allowed for comparison of genotypes with large, medium, and small differences in virulence (exp. 1, 2, and 3 respectively). In experiment 1 and 2, a group of fish was also simultaneously exposed to a mixed infection of each viral genotype at the same dosage as for single infections (4 x $10^5$pfu/ml total virus). Each treatment group initially contained 20 fish, and a negative control group of 8 fish exposed to MEM-10 was also included. After a 1-hour static immersion challenge, flow was resumed to tanks for 1 hour to flush exposure virus, and then each fish was separated into an individual tank containing 1.5L of static water on a tower rack system (Aquatic Ecosystems) to prevent viral transmission between fish. A 1ml water sample was then taken from each isolation tank after distribution of the fish. Water samples were collected from each tank daily for the first 7–10 days, and every 4–7 days thereafter, through a total of 30–35 days. After each sampling the water flow was turned on to all tanks for 1 hour to flush virus, and then tanks were held static for 23 hours until next sampling, as previously described [82,83]. When samples were not collected on successive days the water was left flowing until 23 hours prior to the next sampling, and then water flow was turned to static. As such, each water sample represented the cumulative amount of virus shed during the 23 hours prior to sampling. Water samples were stored at -80˚C until further processing for viral shedding quantification. Fish mortalities were recorded daily, then dead fish were left in their tanks and continued to be sampled for the duration of the experiment (30–35 days).

## Viral shedding quantification

Viral shedding was quantified as previously described [19,64,83]. Briefly, total RNA was extracted from 550ul of daily water samples using the QIAamp MinElute Virus Spin Kit (Qiagen) according to manufacturer's protocols and resuspended in 20ul of nuclease free water. A 11ul volume of eluted RNA was converted to cDNA using random hexamers, oligo-DT, and Moloney murine leukemia virus (MMLV) reverse transcriptase (Promega) in a 20ul reaction, with a 5-minute annealing at 70˚C, followed by a 1-hour reverse transcription at 42˚C, then a 15-minute deactivation at 70˚C. A 5ul volume of cDNA diluted 1:5 in water was then used to quantify viral RNA copy numbers in the water with genotype specific real time quantitative PCR (qPCR) assays specific for each genotype in a 12ul reaction with 40 cycles of 95˚C for 15-seconds and 60˚C for 1-minute. All qPCR primers and TaqMan MGB probes (Applied Biosystems) targeted the viral G gene, the details for which can be found in: [64,67]. In experiment 3, new genotype-specific qPCR assays were developed using forward primers 5'-ACTCCGCT CATTCTCATCTTGAT, 5'-CCACTCCGCTCATTCTCATTCTA; reverse primers 5'-AGAGT GCATCCCTCTGGATAGGT, 5'- CCAGAGTGCATCCCTCGG; and probes 5'-6FAMA CAC TGTAAGTGAATCAGACMGBNFQ, 5'-6FAM ACACCGCAAGTGAGTCAMGBNFQ; for genotypes HV and LR80 respectively. Validation of these qPCR assays was conducted as previously described [64] and all assays allowed for specific quantification of each genotype alone as well as in mixed infections. For each qPCR run, 8-step 10-fold dilution series of RNA transcripts for each IHNV genotype were also included, to allow for absolute quantification of viral RNA copies per ml of water, as previously described [64,67].

## Statistical analyses

All analyses were conducted in R version 3.6.0 [128] using the R studio interface (version 1.0.136, [129]). Unless otherwise noted, each experiment was analyzed separately to avoid any confounding effects. For example, a given IHNV isolate may differ in absolute virulence and shedding between experiments, and this phenomenon was observed here. However, the relatively virulence and shedding differences between isolates is typically consistent across experiments [19,58,67,68]. Furthermore, the experiments were not fully factorial with each-other in

that they had different genotypes, sampling time points, and inclusion of co-infection. Dead fish were also dropped from all analyses from the day after death forward, because previous studies indicate viral transmission rapidly decreases after fish death [83,94]. Inclusion of dead fish in the analyses was explored, but did not change the overall statistical inferences. Mock exposed fish were also dropped, except for qualitative comparisons. Day 0 was dropped from the viral shedding analyses because the goal was to compare treatments from the time all fish began shedding. Best fit models were chosen by exploring all possible factor combinations and the model with the lowest AIC value, which converged, was chosen. If the delta AIC between models was less than 2, the simpler was chosen, according to parsimony. The statistical inferences provided by the best fit model are presented.

Survival was analyzed with Cox proportional hazard models using the coxph function in the Survival package [130], with the proportional hazard assumption validated using cox.zph. Fish that did not die were censored on the last day of the experiment. The number of fish shedding through time was analyzed with generalized linear mixed effects models (glmer function lme4 package [131]), with a binomial distribution and the response variable infection status set to 0 (no detectable shed virus) or 1 (detectable shed virus) for each fish on each day. Fixed explanatory factors included: day (continuous), competition (alone vs. mix), and genotype. The random factor of fish nested within day was also included because the same fish were sampled at multiple time points. The same factors were included in the shedding intensity analyses, which utilized a linear mixed effects model (lme function, nlme package; [132]) with the response variable, daily viral load shed (Log10 (viral genotype RNA copies /ml $H_2O$)) from virus positive fish only. An autoregressive correlation structure (corAR1) and weighted variance structure (varPower) were explored for improving model fit. Model selection was done using maximum likelihood (ML) estimation of parameters. Final coefficients of best fit model were determined using restricted maximum likelihood (REML). Differences between factor levels were determined using least-square means comparison (lsmean) with Tukey's corrections. Normality and autocorrelation of residuals were visually inspected. Because of the exponential change in the quantity of virus shed through time, the total shedding (cumulative amount of virus shed for individual fish up to specified time point) was analyzed in the peak (day 0–4) and post peak period (day 5 –onward) separately. Explanatory factors were genotype and competition, as well as the random factor fish, to account for the two measurements taken in mixed infections (one for each genotype). During the peak period all fish shed detectable virus and the total virus shed data followed a log normal distribution. Therefore, total virus shed was log transformed and analyzed using mixed effects (lme; exp. 1 and 2) or standard linear models (lm; exp. 3, because no mixed infections were assessed). For the post-peak period a large number of fish did not shed virus, so generalized mixed models (glmmTMB; [133]) were used. This made it possible to check for zero inflation in the data. The default unstructured error matrix was utilized and both exponential and linear negative binomial data structures were evaluated.

## Association viral fitness and virulence for individual fish

We sought to elucidate if relationships between virulence and viral fitness might occur on an individual fish basis, which were not visible when looking at treatment mean parameter values. This allowed for direct testing of virulence tradeoff theory and was conducted with data pooled across all experiments to look for general effects [1]. For each infected fish, we quantified virulence as 1/the day of fish death, recovery rate as 1/duration of shedding, transmission rate as peak viral load shed, and total transmission potential as total amount of virus shed (estimates overall virus fitness). Uniform, linear, and quadradic, and exponential functions were fit to the

data and the simplest model was chosen unless a more complex model provided a delta AIC value of 2 or greater. Mixed infections were excluded due to potential confounding effects on virulence. For fish that did not die, virulence was set to 0. The analysis was also run excluding fish for which mortality did not occur, but this did not change the results. To correct for potential impacts of fish death on the duration of shedding, the association between virulence and recovery rate was also examined after excluding fish that died before shedding ending, but this did not alter the findings, so results from all fish are presented.

## Transmission model of IHNV genotypes

In order to infer how IHNV virulence might be selected at a population level on a typical trout farm in North America, a transmission model was developed. This was conducted for genotypes HV and LV, because they were the only genotypes for which appropriate data were available for model parameterization. The model utilized a standard SIRD (susceptible, infected, recovered, dead) framework with the following differential equations:

$$\frac{dS}{dt} = \varphi - \sum_{g \in \{HV, LV\}} \beta_g S I_g - \gamma S \tag{1}$$

$$\frac{dI_g}{dt} = \beta_g S I_g - \rho_g I_g - \alpha_g I_g - \gamma I_g \tag{2}$$

$$\frac{dS}{dt} = \sum_{g \in \{HV, LV\}} \rho_g I_g - \gamma R \tag{3}$$

$$\frac{dD}{dt} = \sum_{g \in \{HV, LV\}} \alpha_g I_g \tag{4}$$

Where parameters $\varphi$ and $\gamma$ equal the stocking and harvest rate of fish, respectively, on a typical trout farm [134,135]. Parameters $\beta_g$, $\rho_g$, and $\alpha_g$ equal the transmission, recovery, and death rate respectively, of fish infected with each index IHNV genotype $g$ (HV or LV). Transmission rate was calculated as $b_g * d_g$, where $b_g$ equals infection rate as a function of dose, and $d_g$ equals virus exposure dose for each viral genotype ($g$). The parameter $b_g$ was derived from McKenney et al. (2016), and transformed to rate fish infected/PFU virus/L water/Day (HV = 6.12 x $10^{-6}$, LV = 1.92 x $10^{-6}$). We assumed that exposure time scales the same as virus concentration, such that a dose of X pfu total virus after 1-hour exposure scales to a dose of 24*X pfu total virus exposure after a 24-hour exposure. Data from experiment 1 was used to parameterize $d_g$ with the equation $d_g$ = *mean shedding rate/volume typical raceway in a fish farm/water exchange rate in a fish farm*. Mean shedding rate = total pfu virus shed/number days shedding, averaged across each fish. It was assumed that 1 viral RNA copy shed is equivalent to 1 PFU. Although this relationship has not been established, it is unlikely to change the relative differences between genotypes and only speed up or slow down overall transmission dynamics. The raceway volume (183,219 L), and water exchange rate (72 flushes/day), were chosen to represent conditions on a typical North American trout farm [134–137]. Transmission is also assumed to be driven by fish contact with virus particles in the water [69–75,82–84]. Recovery rate equaled 1/mean duration of shedding (last day-first day + 1) for fish infected with each IHNV genotype; calculated using Kaplan-Meier analysis (survfit) on experiment 1 data. Fish that died before shedding ended, were censored on the day of death. Death rate was 1/mean time to death; calculated using Kaplan-Meier analysis (survfit) on experiment 1. Fish that did not die

were censored on the last day of the experiment. The impact of culling, i.e. removal of all fish from a production raceway in response to a disease epidemic, on IHNV virulence selection was also investigated. Culling was triggered in the model if the total number of dead fish was equal to or greater than 30% of the original population size. If culling occurred, S, I, R, and D were all set to zero in the model simulation, regardless of infection status. How infection of fish populations with a single IHNV genotype versus simultaneously with both genotypes, influenced virulence selection, was also investigated. It is important to note that although fish populations could be simultaneously infected with multiple genotypes, individual fish could only be infected with a single genotype. This simpler model framework was chosen because experimental results indicated that mixed infection dynamics did not substantially alter the relative differences between genotypes (see results section). The transmission model was run using the deSolve library in R [138], with state variables updated once per day (smaller intervals were examined and did not change results). The goal of the transmission modelling was to investigate relative differences between genotypes in relation to virulence evolution, with field relevance, rather than provide absolute transmission dynamics on a specific trout farm.

## Supporting information

**S1 Text.** Figure A shows individual fish viral shedding profiles and Tables A-L shows selected minimal models from statistical analyses.
(DOCX)

## Acknowledgments

We would like to thank Alison Kell, Maureen Purcell, Rachel Breyta, Douglas McKenney, David Kennedy, David Paez, and Jim Winton for valuable discussions. We are grateful to Scott LaPatra and Randy MacMillan of Clear Springs Foods Inc. for providing research grade rainbow trout. Any use of any trade, firm, or product names is for descriptive purposes only and does not imply endorsement by the U.S. Government.

## Author Contributions

**Conceptualization:** Andrew R. Wargo, Gael Kurath, Benjamin Kerr.

**Data curation:** Andrew R. Wargo, Robert J. Scott.

**Formal analysis:** Andrew R. Wargo, Gael Kurath, Benjamin Kerr.

**Funding acquisition:** Andrew R. Wargo, Gael Kurath, Benjamin Kerr.

**Investigation:** Andrew R. Wargo, Robert J. Scott.

**Methodology:** Andrew R. Wargo, Gael Kurath, Robert J. Scott.

**Project administration:** Andrew R. Wargo, Gael Kurath.

**Resources:** Andrew R. Wargo, Gael Kurath.

**Software:** Andrew R. Wargo, Benjamin Kerr.

**Supervision:** Andrew R. Wargo, Gael Kurath, Benjamin Kerr.

**Validation:** Andrew R. Wargo, Gael Kurath.

**Visualization:** Andrew R. Wargo, Gael Kurath, Benjamin Kerr.

**Writing – original draft:** Andrew R. Wargo.

**Writing – review & editing:** Andrew R. Wargo, Gael Kurath, Benjamin Kerr.

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
