## [Decision Letter · Decision Letter 0]

29 Jan 2021

Dear Dr. Wargo,

Thank you very much for submitting your manuscript "Virus shedding kinetics and unconventional virulence tradeoffs" for consideration at PLOS Pathogens. As with all papers reviewed by the journal, your manuscript was reviewed by members of the editorial board and by several independent reviewers. The reviewers appreciated the attention to an important topic. Based on the reviews, we are likely to accept this manuscript for publication, providing that you modify the manuscript according to the review recommendations.

Sincerely,

Marco Vignuzzi

Section Editor

PLOS Pathogens

Kasturi Haldar

Editor-in-Chief

PLOS Pathogens

orcid.org/0000-0001-5065-158X

Michael Malim

Editor-in-Chief

PLOS Pathogens

orcid.org/0000-0002-7699-2064

Reviewer Comments (if any, and for reference):

Reviewer's Responses to Questions

**Part I - Summary**

Reviewer #1: In this study, the authors carry out a thorough and impressive study to test the virulence-transmission trade-off hypothesis. This hypothesis has formed the foundation of a wealth of infectious disease models, predicting that virulence can evolve as an unavoidable consequence of natural selection acting on between-host transmission. While aspects of this theory have been supported in multiple systems, very few studies have demonstrated the non-linear relationships between virulence, transmission and recovery and the truncation of transmission (the real cost of virulence) that underlie the theory. As such, empirical tests of the theory remain badly needed.

In the first part of this manuscript, the authors describe three experiments in which they infected rainbow trout with different genotypes of infectious hematopoietic necrosis virus (IHNV), then measured virulence (host mortality), recovery (viral titers going below detection levels) and transmission rate (viral particles released into wat column) over the ensuing course of infection. They used a thorough sampling scheme, allowing them to calculate lifetime transmission and to assess the relationships between the different variables. In addition to studying single-genotype infections, they also included mixed infections in two experiments, allowing them to study competitive interactions between viral strains. Overall, the authors do not find support for the trade-off hypothesis: while more virulent viruses obtain greater transmission, the truncation of the transmissible period due to death and recovery were not strong enough to put a break on increasing virulence. Thus, viral fitness was not maximized at an intermediate level of virulence, but instead kept rising with increasing virulence. This is an important finding. While negative results can sometimes potentially be ascribed to studies that have low replication or that do no thoroughly measure different fitness aspects, I feel that the current study is designed well and thoroughly enough to warrant the conclusion that there is no support for the trade-off hypothesis in this system.

A remaining question then is what maintains variation in virulence, and what can put a break on increasing virulence? To address this, the authors describe a mathematical model in the second part of their manuscript. In this model, based on a typical SIR-type model, they release viral strains of varying virulence in a virtual agricultural trout setting, and then track the spread of the strains. Interestingly, the model shows that culling – where all fish in an operation are culled based on a threshold outbreak size, as happens in real operations – provides the break on increasing virulence: the strain with lower virulence becomes the dominant strain in such operations. I think this is an interesting finding that provides novel insights in the effects if culling (these effects are not always intuitive) and provides an additional mechanism (the literature often ascribes the maintenance of virulence variation to GxG interactions or population heterogeneity) for the maintenance of variation in virulence.

One more impressive aspect of this manuscript is the thorough way in which the literature is cited and summarized: indeed, the manuscript would provide a valuable introduction to virulence evolution for a broad group of scientists.

Reviewer #2: This is a very interesting manuscript addressing virulence evolution in a well-characterized fish-virus system. The experiments are well done and the questions asked are interesting and significant. The manuscript addresses several aspects of virulence evolution (trade-off theory, co-infection, effects of culling), which I think ends up being both a strength and a weakness. It is a strength in that there are several different potential aspects of virulence evolution addressed. The challenge is that there are so many different components to the study that it is a challenge to adequately set them up in the introduction and interpret the results in the discussion clearly. Most of my comments center around this challenge.

Reviewer #3: In their manuscript Wargo et al. investigate how viral transmission rate and duration are associated with virulence in relation to the trade-off theory, using in vivo studies including infectious hematopoietic necrosis virus (IHNV) isolates with varying virulence to infect its natural host, rainbow trout (Oncorhynchus mykiss). Three experimental challenges were performed, each including two IHNV genotypes in varying combinations of known high/low virulence. A total population of 20 fish per genotype was used, but each fish was handled and studied individually in separate systems.

The study is thorough and comprehensive, seems well planned, is very well written and in total creates a convincing basis for the conclusions. It is also based on several related studies from the same authors and others and in general on a well documented fish virus and infection system on trout. The study is recommended for publication in Plos Pathogens after corrections as described below.

RESULTS chapter

P9 - “Number of fish shedding”-section and P54 - Fig 2

Materials and methods chapter (P26L561) described that dead fish were left in their tanks and that water samples were continued to be taken for the full duration of the experiment. While reading the “Number of fish shedding”- section, I was starting to wonder how these results was (or wasn’t) included in e.g. measurements of viral shedding. I eventually found in figure legends that these results were only included using the sampling at 24 hours after death. I suggest some information in the text part as well on which samplings (i.e. live fish only) the results are based on.

Regarding Fig. 2 and e.g. statements in line 180-182 (“Shedding then gradually decreased and by day 15 the majority of fish had stopped shedding. However, a small number of fish (1-5), were still shedding detectable virus at the end of each experiment (day 30-35).”): I started to question myself if “the majority of fish” weren’t dead by day 15 (at least for some isolates)? I suggest it could be clearer if the “Number of fish” in Fig 2 and “majority of fish” in text were replaced and presented to relate to the remaining live fish only (e.g. “Percent of live fish” or similar). The reader will get an impression on how many live fish there are left at each time points from fig 1, but it is hard to relate that precisely to fig 2 and results description. “Percent live fish” is used in an example mentioned in Discussion chapter (P17L366-369) and I would like similar presentation to be used in Fig. 2 and related text as well.

P10 – “Shedding intensity”-section

Similar as described above, is also related to this section where line 210-211 states that “Afterwards shedding intensity remained stable, although the number of fish shedding continued to decrease”. Please relate/correct this to relate to the number of live fish.

In Fig 3 legends it is stated (L245-246) that “Number of fish included in the mean values can be found in figure 2.”, but I cannot easily find such information in Fig 2 (but would definitely like to have it).

P12 – “Total virus shed”-section

L249 “..the total amount of virus shed by all 20 fish in each treatment group over the course of infection” - Please specify how this relates to live fish as it can be interpreted that you include the amount “shed” (i.e. released) from decomposed fish tissue of dead fish (available data on this is shown in fig S1.)

MATERIALS AND METHODS chapter

A lot of the methodology is well described previously, and the authors refer to previous publications describing it. This is good and increase quality as the methods have been used earlier, but for the reader it would be easier if also a SHORT description of the basics was included, where applicable. Please consider.

SUPPLEMENTARY

Mat & Met chapter P26 and Fig S1: Mat & Met describes that “dead fish were left in their tanks and continued to be sampled for the duration of the experiments (30-35 days)”. The result of these analyses of virus presence in the water is shown in fig S1, but is not commented or discussed anywhere. I would like to know the intention of including this and also that the results were commented in the manuscript. It seems that (by a rough overview) e.g. that fish infected with high virulent isolates release virus for quite a long time post mortem (probably to a large extent due to decomposing of tissue as it can not be defined as true “shedding” due to replicating virus). These results are probably not relevant for the discussion of trade-off theory in the manuscript, but if included it is worth a comment.

**Part II – Major Issues: Key Experiments Required for Acceptance**

Reviewer #1: none

Reviewer #2: The experiments described, and their conclusions are robust. My only major suggestion is with respect to the statistical analyses. I understand why the authors chose to analyze each experiment separately, but also felt that this made it very hard to draw general conclusions. Given that the experiments were all done identically, could some of the analyses (with respect to fish shedding, for example) be done as one overall model, treating virulence as a continuous trait to account for the fact that you had distinct levels and differences in virulence across the experiments? That is essentially what you did in the individual-level analysis of virulence trade-offs, but it seems to me like it could also be done as a strain-level analysis. Perhaps you would still want to examine the effects of coinfection separately for each "experiment" but it seems as though it would make for a stronger analysis (and clearer conclusions) if many of your other results could be part of one larger analysis that crosses experiments and strains. See editorial comments below- but if you did change the analysis accordingly, I would suggest separating out the question about mixed infections as its own section of results that is distinct from the trade-off questions. I found it very hard to follow all the various results in the current format, which detracted from their importance.

I also had one question about the interpretation of mortality data from the experiments. To what extent do mortality rates in your experiments reflect mortality in "nature" (or aquaculture)? Is it possible that mortality rates are significantly higher outside of a lab environment, reducing the advantage of the shedding duration for virulent strains? That wouldn't really alter your results much given how much of the shedding occurs before mortality, but I wondered about the extent to which these parameters would extrapolate to field settings. Is it possible that more mortality occurs earlier in the field, for example, due to predation of sick fish? Any evidence that fish behavior changes during acute shedding in a way that may make fish more prone to predation?

Reviewer #3: None

**Part III – Minor Issues: Editorial and Data Presentation Modifications**

Reviewer #1: 1. Results: when reading the results I wondered why the data were not analyzed while excluding the zeroes (fish that did not die). This is later mentioned in the methods, but I think it would be worth mentioning this in the results already, and potentially showing best fit relationships for both sets of data (with and without fish that did not die).

2. Lines 298-299. When reading this section, I wondered why only an analysis of individual fish was presented. What would relationships across the experiments look like if data were analyzed on a per-genotype basis?

3. Line 359: were trade-offs really observed? Or were positive relationships observed? Trade-offs are really negative relationships. In the trade-off hypothesis, the trade-off arises because virulence increases transmission rate while reducing the transmissible period.

Reviewer #2: Editorial suggestions:

The manuscript addresses a number of different questions using the same set of experiments, including mixed infections and their effects on virulence. I didn't see any mention of mixed infections (or culling effects) in the introduction, which made it feel a bit out of nowhere when the results were presented. Could some rationale / context for those aspects of the study be incorporated into the introduction? Relatedly, given the format that PLoS uses with the methods at the end, a few more details about the experiments should be presented either at the start of the results or the end of the intro (in particular, I was wondering whether the dose used in mixed infections was identical to those for single infections, so that seems like a key detail to have before the results are presented). I also had trouble figuring out the motivation for the mixed infection studies given that they are thought to be uncommon in the field. Perhaps this could be set up in the intro.

In the discussion, I wondered if the authors might be able to speculate about why some systems might conform to the classical trade-off model but not others. This may not be something they feel comfortable doing at this point, but I think the results are important and I wondered if there are certain characteristics of the system (water-borne pathogen, etc.) that may make it more likely to fall outside of the traditional trade-off model?

Line 51: a word "in" is missing here after "resulting".

Line 78: This may be personal preference but this paragraph is really long. Might help to start a paragraph break at "In recent years."

Line 110: an "of" is missing after "fitness traits"

Line 186: again this may be personal preference, but I found the phrasing awkward - essentially a double negative. Could you say instead "with a more rapid decline in probability of shedding for the less virulent genotype..."

Line 249: I assume you mean "on average" when you say "by all 20 fish"? I would clarify because when I first read it I thought that you had summed the data across all 20 fish.

Line 255: minor but I don't think you need the words "the data analysis revealed" since it's implied

Lines 261-263: it took me awhile to figure out how this result differed from the one presented above. This section could benefit from adding extra text to clarify the distinction between the various questions asked. I wondered if perhaps even separating out the mixed infection questions from the virulence questions would help structure the results better. I struggled to follow all of the various results.

Line 292: is "transmission rate" here akin to shedding? I think you define it later but I would be clear upon first use about what you mean.

Line 298: I would add "viral" before fitness both at the end of line 298 and in the next sentence in line 299, just to clarify that it is viral rather than fish fitness you are referring to. Similarly for line 311.

Line 315. "parameters" should be "parameterized"

Line 336: personal preference but I suspect adding a comma after "culling" for clarity

Lines 545-548: this all seem like key details for interpretation of the results- could they be moved to to the start of the results?

Line 627: should "at" be "as"?

Line 638: I think "from" is meant to say "farm"

Figure 1. The panels are very small- this may just be the case for review but they are difficult to read at the current size. I wonder if you could use color coding that better corresponds to strain virulence somehow, to make interpretation of the graphs a bit easier.

Figure 5. Given that these lines are not statistically significant, I would make them dashed instead of solid.

Reviewer #3: P14L212 and P11L224: Suggest also to refer to fig 3 in addition to Tables.

P15L326 and 328: Please check if correct fig references are used

P18L401: Please check if correct fig references are used (e.g. there are no Fig 4D in the manuscript)

A few proofreading corrections are still needed.

PLOS authors have the option to publish the peer review history of their article (what does this mean?). If published, this will include your full peer review and any attached files.

Reviewer #1: No

Reviewer #2: No

Reviewer #3: No
---

## [Editor Report · Decision Letter 1]

3 Apr 2021

Dear Dr. Wargo,

We are pleased to inform you that your manuscript 'Virus shedding kinetics and unconventional virulence tradeoffs' has been provisionally accepted for publication in PLOS Pathogens.

Best regards,

Marco Vignuzzi

Section Editor

PLOS Pathogens

Kasturi Haldar

Editor-in-Chief

PLOS Pathogens

orcid.org/0000-0001-5065-158X

Michael Malim

Editor-in-Chief

PLOS Pathogens

orcid.org/0000-0002-7699-2064

Reviewer Comments (if any, and for reference):

No additional comments

---

## [Editor Report · Acceptance letter]

27 Apr 2021

Dear Dr. Wargo,

We are delighted to inform you that your manuscript, "Virus shedding kinetics and unconventional virulence tradeoffs," has been formally accepted for publication in PLOS Pathogens.

Best regards,

Kasturi Haldar

Editor-in-Chief

PLOS Pathogens

orcid.org/0000-0001-5065-158X

Michael Malim

Editor-in-Chief

PLOS Pathogens

orcid.org/0000-0002-7699-2064